**Brief Communication**

# Structural basis of the histone ubiquitination read–write mechanism of RYBP–PRC1

Maria Ciapponi[1], Elena Karlukova[1], Sven Schkölziger[1], Christian Benda [2]✉ & Jürg Müller [1]✉

Histone H2A monoubiquitination (H2Aub1) by the PRC1 subunit RING1B entails a positive feedback loop, mediated by the RING1B-interacting protein RYBP. We uncover that human RYBP–PRC1 binds unmodified nucleosomes via RING1B but H2Aub1-modified nucleosomes via RYBP. RYBP interactions with both ubiquitin and the nucleosome acidic patch create the high binding affinity that favors RYBP- over RING1B-directed PRC1 binding to H2Aub1-modified nucleosomes; this enables RING1B to monoubiquitinate H2A in neighboring unmodified nucleosomes.

Monoubiquitination of histone H2A at lysine 119 (H2Aub1) is catalyzed by the RING1B subunit of the E3 ubiquitin ligase Polycomb Repressive Complex 1 (PRC1)[1,2]. To ubiquitinate nucleosomes, the N-terminal ring finger domain of RING1B, which enables ubiquitin transfer from E2 to H2A, must associate with the N-terminal ring finger domain of the PRC1 subunit BMI1 (refs. 3–5). RING1B is also the common core subunit of variant forms of PRC1 (vPRC1)[6,7]. In vPRC1, also called RYBP–PRC1, the RING1B C-terminal RAWUL domain is bound by the RING1B-interacting domain of RYBP or its paralogue YAF2, whereas in canonical PRC1, the RAWUL domain is associated with CBX proteins[6–10]. For a recent review of the composition and function of different PRC1 forms, see ref. 11. The findings that RYBP contains an Npl4-type zinc finger (NZF) that binds ubiquitin[12] and that RYBP is a principal interactor of H2Aub1 nucleosomes[13] provided an indication that RYBP binding to H2Aub1 may be part of a positive feedback loop that facilitates formation of H2Aub1-modified chromatin domains. Supporting this, depletion of Rybp and Yaf2 in mouse embryonic stem cells (ESCs) greatly reduced H2Aub1 levels in such domains at PRC1 target genes[14]. Likewise, RYBP–PRC1 could interact with the ubiquitin moiety on preinstalled H2Aub1 to efficiently promote H2A monoubiquitination on juxtaposed unmodified nucleosomes in recombinant nucleosome arrays[15]. Here, single-particle cryo-electron microscopy (cryo-EM) uncovers that RYBP–PRC1 uses two different interfaces for contacting unmodified and H2Aub1-modified nucleosomes. These distinct binding modes enable the complex to bind to H2Aub1 and simultaneously monoubiquitinate unmodified H2A via a read–write mechanism.

We reconstituted and purified a RYBP–PRC1 complex containing full-length human RING1B, BMI1 and RYBP (Fig. 1a,b). We shall refer to this minimal complex as vPRC1. For cryo-EM analysis, vPRC1 was mixed with 5′ biotinylated *Drosophila melanogaster* mononucleosomes that were either unmodified (Nuc) or fully monoubiquitylated on H2A (Nuc$_{H2Aub1}$) (Fig. 1c) and each sample was bound separately to grids coated with streptavidin. We shall present the results from the single-particle analysis of vPRC1:Nuc and vPRC1:Nuc$_{H2Aub1}$ in turn.

The vPRC1:Nuc sample showed a homogeneous population of two-dimensional (2D) classes. Three-dimensional (3D) reconstruction to a resolution of ~2.9 Å revealed nucleosomes with one RING1B:BMI1 ring finger heterodimer (RING1B$_{15–115}$:BMI1$_{3–107}$) bound to each face of the nucleosome disc (Fig. 1a,d, Extended Data Fig. 1 and Table 1). We could not detect density for the RING1B or BMI1 C termini, or for RYBP. The structure of these vPRC1:Nuc particles is comparable to that of the minimal RING1B:BMI1 ring finger-dimer bound to the nucleosome core particle, determined previously by crystallography[5] (Extended Data Fig. 2). In brief, RING1B contacts the H2A acidic patch residues E60, E63, D71, N88, D89 and E91 within the nucleosome, whereas BMI1 interacts with H4 and H3 residues, as reported[5] (Fig. 1e and Extended Data Figs. 1 and 2).

The vPRC1:Nuc$_{H2Aub1}$ sample also showed homogenous 2D classes that were distinct from those obtained with the vPRC1:Nuc sample, as they lacked the characteristic density for RING1B or BMI1. Refinement to a final resolution of ~3.18 Å revealed symmetrical density on both faces of the nucleosome disc (Extended Data Fig. 3). By focusing refinement on one nucleosome surface, we obtained an improved map with partial side chain information that allowed us to assign the extra density to a fragment of RYBP (RYBP$_{23–58}$) and ubiquitin (Fig. 1a,f–h, Extended Data Fig. 3 and Table 1). The resulting model shows RYBP NZF residues T31, F32 and I43 in contact with ubiquitin I44—an interaction that closely resembles that of the related Npl4 zinc finger with

[1]Laboratory of Chromatin Biology, Max-Planck Institute of Biochemistry, Martinsried, Germany. [2]Department of Structural Cell Biology, Max-Planck Institute of Biochemistry, Martinsried, Germany. ✉e-mail: benda@biochem.mpg.de; muellerj@biochem.mpg.de

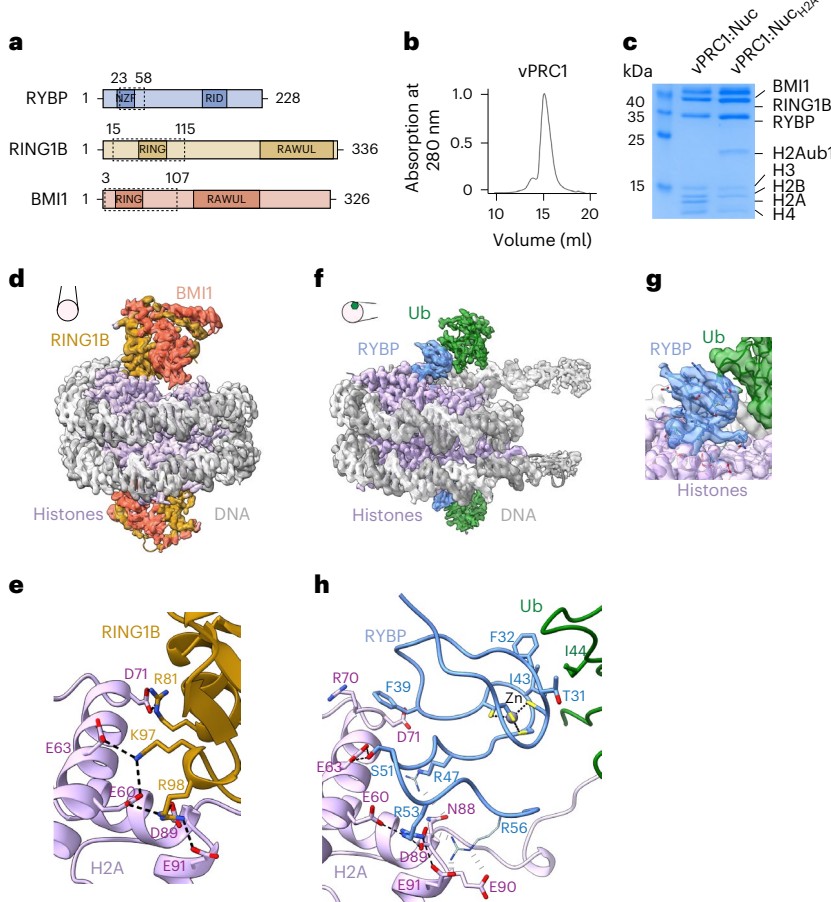

**Fig. 1 | vPRC1 binds unmodified nucleosomes via RING1B:BMI1 and H2Aub1-modified nucleosomes via RYBP. a**, Domain organization of the vPRC1 subunits with the NZF and Ring1b-interacting domain (RID) in RYBP, and the ring finger (RING) and RAWUL domains in RING1B and BMI1. Dashed boxes indicate protein regions built into the models. **b**, Gel filtration profile of reconstituted vPRC1 used for cryo-EM. **c**, Coomassie-stained SDS–PAGE showing vPRC1:Nuc and vPRC1:Nuc$_{H2Aub1}$ samples used for cryo-EM. **d**, Cryo-EM reconstruction at 2.9 Å and final model of vPRC1:Nuc showing the RING:BMI1 ring finger heterodimer (gold and orange) bound to each face of the nucleosome disc (gray, DNA; pink, histone octamers). View orientation of the nucleosome relative to its dyad axis is indicated. **e**, Interaction between RING1B and the H2A acidic patch; the interacting residues are represented as sticks and contacts are marked by dashed lines. **f**, Cryo-EM reconstruction at 3.18 Å and model of vPRC1:Nuc$_{H2Aub1}$ showing RYBP$_{23–58}$ (blue) and ubiquitin (green) bound to each face of the nucleosome disc. View orientation of the nucleosome relative to its dyad axis is indicated. **g**, Focused refined map showing density of the RYBP zinc finger contacting ubiquitin and RYBP loop binding the nucleosome acidic patch. **h**, Interactions of RYBP$_{23–58}$ with the H2A acidic patch and ubiquitin. Contacts between sidechains with clear density are represented as stick models and black dashed lines. Interactions with weak or unresolved density are indicated in gray and sidechains in pale colors. Acidic patch residues E60, E63, D70, N88, D89 and E91 are contacted by RYBP, whereas in **e** the same residues are contacted by RING1B.

ubiquitin (Fig. 1g,h)[16]. The RYBP residues 47–58 adjacent to the NZF form an extended loop across the entire H2A acidic patch (Fig. 1g,h and Extended Data Fig. 3b,e). Central to this interaction are RYBP residues R47, R53 and R56. The side chain of RYBP R53 contacts H2A residues E60, D89 and E91 (Extended Data Fig. 3e). For RYBP residues R47 and R56, the side chain density was less well defined but their guanidino groups probably form additional contacts with the H2A acidic patch residues N88, and D89, E90 and E91, respectively (Fig. 1h and Extended Data Fig. 3e). Moreover, RYBP F39, showing clear side chain density, interacts with H2A residues R70 and D71 (Extended Data Fig. 3e). These structural studies show that vPRC1 binds unmodified nucleosomes via RING1B:BMI1. However, they also reveal a previously uncharacterized binding mode to H2Aub1-modified nucleosomes via RYBP. The observation that, in these two binding modes, vPRC1 occupies the same acidic patch surface argues that these two binding interactions are mutually exclusive.

We next investigated how RYBP directs vPRC1 binding to chromatin by performing electromobility shift assays (EMSA) on mononucleosomes. Reconstituted recombinant mononucleosomes were assembled on a 3′-ATTO647N-labeled 201-base pair (bp) DNA fragment containing the 601 nucleosome-positioning sequence, and they were either unmodified or contained H2Aub1.

First, we analyzed how ubiquitin on H2A affects the binding of full-length RYBP or the RING1B:BMI1 heterodimer to nucleosomes. To detect changes in nucleosome mobility in EMSAs upon binding of RYBP, we increased the molecular weight of RYBP by fusing it to mNeonGreen—a protein lacking nucleosome-binding activity (Extended Data Fig. 4a). The mNeonGreen-RYBP fusion protein (NG–RYBP) bound H2Aub1 nucleosomes with almost tenfold higher affinity compared with the RING1B:BMI1 dimer (Fig. 2a). Of note, H2Aub1 does not appear to impact on RING1B:BMI1 binding to nucleosomes, as the RING1B:BMI1 dimer bound H2Aub1-modified and unmodified nucleosomes with similar affinity (Fig. 2a). In contrast, NG–RYBP bound to H2Aub1-modified nucleosomes with almost fourfold higher affinity than to unmodified mononucleosomes (Extended Data Fig. 4b).

We next tested the nucleosome-binding affinity of NG–RYBP with mutations in residues contacting ubiquitin (NG–RYBP$_{T31A/F32A}$) or those

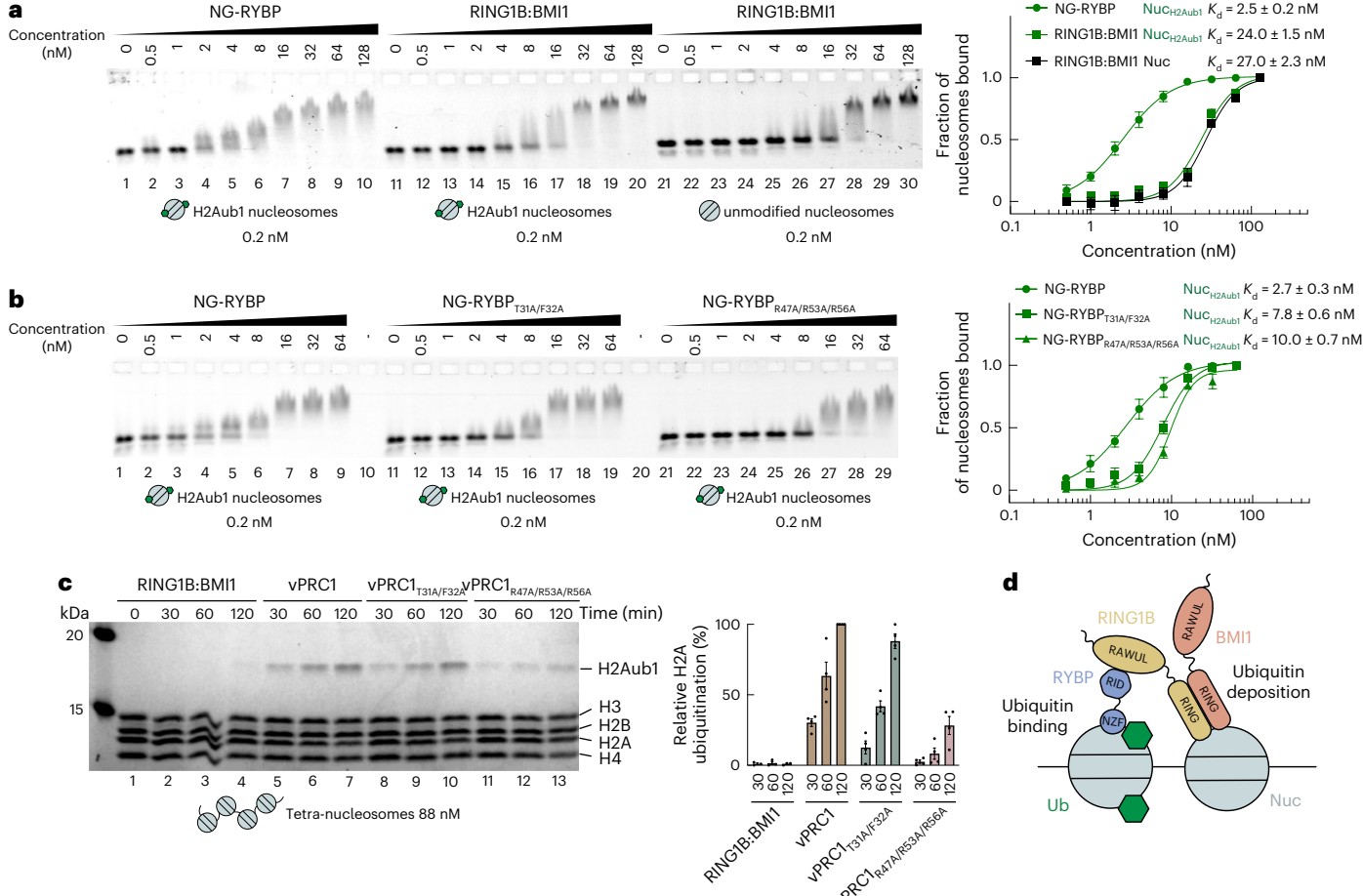

**Fig. 2 | High-affinity binding of RYBP to H2Aub1 nucleosomes via ubiquitin- and acidic patch contacts enables vPRC1 to generate H2Aub1 chromatin domains.** **a**, RYBP shows higher binding affinity than RING1B:BMI1 on H2Aub1 nucleosomes. Left, EMSA with the indicated concentrations of NG–RYBP or RING1B:BMI1 and 0.2 nM 647N-ATTO-labeled mononucleosomes that were H2Aub1-modified or unmodified as indicated. Right, quantitative analysis of EMSA data by densitometry of 647N-ATTO signal from independent experiments ($n = 5$), error bars (s.e.m.); apparent dissociation constant ($K_d$) values are indicated. **b**, Mutation of ubiquitin-contacting or acidic patch-contacting residues in RYBP reduces binding to H2Aub1 nucleosomes. Left, EMSA as in **a**, comparing binding of NG–RYBP, NG–RYBP$_{T31A/F32A}$ and NG–RYBP$_{R47A/R53A/R56A}$ with H2Aub1-modified

mononucleosomes. Right, quantitative analysis from independent experiments ($n = 3$) as in **a**. **c**, Efficient H2A monoubiquitination by vPRC1 in nucleosome arrays relies on RYBP interaction with the nucleosome acidic patch. Left, ubiquitination reactions monitoring H2Aub1 formation by full-length RING1B:BMI1, wild-type vPRC1, vPRC1$_{T31A/F32A}$ or vPRC1$_{R47A/R53A/R56A}$ on tetranucleosomes after indicated incubation times, analyzed on Coomassie-stained 16% polyacrylamide gel. Right, quantification of H2Aub1 signal by densitometry from independent experiments ($n = 4$). In each experiment, the H2Aub1 signal in lane 7 was defined as 100% and used for quantification of H2Aub1 signals in other lanes on the same gel. Circles show individual datapoints with error bars (s.e.m.). **d**, Model of the vPRC1 read–write mechanism. Abbreviations as in Fig. 1a.

contacting the acidic patch (NG–RYBP$_{R47A/R53A/R56A}$). Compared with wild-type NG–RYBP, both mutants bound H2Aub1-modified mononucleosomes with three- and fourfold lower affinity (Fig. 2b). The residual binding by NG–RYBP$_{R47A/R53A/R56A}$ at high concentrations (Fig. 2b, lanes 27–29) is not due to interaction with ubiquitin because similar binding occurs on unmodified mononucleosomes (Extended Data Fig. 4c); the nature of this RYBP-nucleosome interaction is currently unknown. Together, these analyses validate our structural data and show that interaction of the RYBP NZF with ubiquitin and of the RYBP loop with the acidic patch are critical for high-affinity binding of RYBP to H2Aub1-modified nucleosomes.

We then compared nucleosome-binding and H2A ubiquitination activity of wild-type vPRC1 complex with that of vPRC1$_{T31A/F32A}$ and vPRC1$_{R47A/R53A/R56A}$ complexes containing the mutant RYBP proteins (Extended Data Fig. 5 and Fig. 2c). Wild-type vPRC1 bound H2Aub1 mononucleosomes with higher affinity than unmodified mononucleosomes, whereas vPRC1$_{T31A/F32A}$ and vPRC1$_{R47A/R53A/R56A}$ did not show this higher binding affinity on H2Aub1 mononucleosomes (Extended Data Fig. 5c,d). The RYBP interactions with ubiquitin and the acidic patch

observed in the structure are therefore both critical for high-affinity binding of vPRC1 to H2Aub1 nucleosomes.

Finally, we assessed vPRC1 activity on tetranucleosome arrays by monitoring the kinetics of H2Aub1 formation. We rationalized that, upon monoubiquitination of H2A on a first nucleosome in an array, RYBP–H2Aub1 contacts might facilitate monoubiquitination of neighboring nucleosomes[15]. Indeed, wild-type vPRC1 generated H2Aub1 more effectively than RING1B:BMI1 dimer alone (Fig. 2c). Mutation of the RYBP loop severely compromised the ability of the vPRC1$_{R47A/R53A/R56A}$ complex to generate H2Aub1 (Fig. 2c), whereas vPRC1$_{T31A/F32A}$ showed reduced H2Aub1 deposition only early in the reaction (Fig. 2c). We conclude that RYBP contacts with ubiquitin and the nucleosome acidic patch are functionally important for efficient formation of H2Aub1-modified chromatin domains by vPRC1.

Positive feedback loop mechanisms where a histone-modifying enzyme uses a 'read–write' mechanism to both bind to and generate a posttranslational modification have emerged as a basic principle for formation of extended chromatin domains carrying such a modification[17,18]. The work here uncovers the molecular basis of how vPRC1

**Table 1 | Cryo-EM data collection, refinement and validation statistics**

| | vPRC1:Nuc (EMD: 17797; PDB: 8PP7) | vPRC1:Nuc_{H2Aub1} (EMD: 17796; PDB: 8PP6) |
|---|---|---|
| **Data collection and processing** | | |
| Magnification | 105,000 | 81,000 |
| Voltage (kV) | 300 | 300 |
| Electron exposure (e⁻/Å²) | 58.6 | 54.0 |
| Defocus range (μm) | 0.5–3 | 0.5–3 |
| Pixel size (Å) | 0.8512 | 1.0940 |
| Symmetry imposed | C1 | C1 |
| Initial particle images (no.) | 6,723,019 | 12,150,042 |
| Final particle images (no.) | 146,136 | 640,334 |
| Map resolution (Å) | 2.91 | 3.18 |
| FSC threshold | 0.143 | 0.143 |
| Map resolution range (Å) | 2.0–7.0 | 2.5–7.0 |
| **Refinement** | | |
| Initial model used (PDB code) | 6PWE, 4R8P | 6PWE, 1Q5W |
| Model resolution (Å) | 2.9 (unmasked) | 3.0 (unmasked) |
| FSC threshold | 0.5 | 0.5 |
| Model resolution range (Å) | n/a. | n/a. |
| Map sharpening B factor (Å²) | −60 | −60 |
| Model composition | | |
| Nonhydrogen atoms | 15,562 | 13,317 |
| Residues | Protein: 1,165 Nucleotide: 306 | Protein: 872 Nucleotide: 312 |
| Ligands | Zn: 8 | Zn: 1 |
| B factors (Å²) | | |
| Protein | 25.13 | 42.62 |
| Nucleotide | 52.00 | 69.40 |
| Ligand | 89.39 | 178.81 |
| R.m.s. deviations | | |
| Bond lengths (Å) | 0.004 | 0.005 |
| Bond angles (°) | 0.598 | 0.562 |
| **Validation** | | |
| MolProbity score | 1.18 | 1.17 |
| Clashscore | 2.40 | 3.48 |
| Rotamer outliers (%) | 0.69 | 0.00 |
| Ramachandran plot | | |
| Favored (%) | 97.11 | 97.89 |
| Allowed (%) | 2.98 | 2.11 |
| Disallowed (%) | 0.00 | 0.00 |

binds H2Aub1-modified nucleosomes and how this binding enables H2Aub1 deposition by the same enzyme molecule on neighboring nucleosomes. Our results argue that the high binding affinity created by the combined interactions of the RYBP zinc finger with ubiquitin and of the RYBP_{47–58} loop with the nucleosome acidic patch directs vPRC1 to dock on H2Aub1-modified nucleosomes. This binding geometry allows the freely exposed RING1B:BMI1 heterodimer in the same vPRC1 molecule to engage with the acidic patch of an unmodified nucleosome in the vicinity and thereby enable monoubiquitination of H2A on this nucleosome (Fig. 2d). There is currently no evidence that vPRC1 binding to an H2Aub1-modified nucleosome would induce a conformational change in vPRC1 to allosterically activate its E3 ligase activity. It rather appears that vPRC1 generates domains of H2Aub1-modified chromatin[14,15] via a simple read–write mechanism.

## Online content

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

## Methods

### Protein expression constructs

Human RING1B (Uniprot: Q99496) and BMI1 (Uniprot: P35226) full-length coding sequences were cloned into the same pFastBac (pFB) vector for coexpression in insect cells. This vector encodes an HRV3C-cleavable N-terminal twin strep-tagg and a C-terminal non-cleavable 6×His tag linked to the RING1B and BIM1 coding sequences, respectively. The human RYBP (Uniprot: Q8N488) full-length coding sequence was cloned into pFB downstream of a HRV3C-cleavable N-terminal 6×His tag. For the mNeonGreen-RYBP (NG–RYBP) expression construct, the *Escherichia coli* expression vector encoding an HRV3C-cleavable N-terminal 6×His-tagged linked to the mNeonGreen coding sequence[19] was modified by fusion of the RYBP codons$_{1–229}$ at its C terminus, separated by a seven-amino-acid codon GSAAAGS linker. RYBP and NG–RYBP mutant expression constructs (RYBP$_{T31A/F32A}$, RYBP$_{R47A/R53A/R56A}$, NG–RYBP$_{T31A/F32A}$ and NG–RYBP$_{R47A/R53A/R56A}$) were prepared by standard site-directed mutagenesis. Detailed plasmid maps are available on request.

### Protein expression and purification

RYBP and the RING1B:BMI1 heterodimer were expressed separately in insect cells[20]. An optimized ratio of the baculoviruses (produced in Sf21 cells (Invitrogen, cat. no. 1149701)) for the different vPRC1 subunits was used to infect *Trichoplusia ni* High Five insect cells (Invitrogen, cat. no. B85502). Cells were lysed using a glass Dounce homogenizer and proteins were purified using Ni-NTA affinity chromatography, followed by His or twin strep-tag cleavage for His-RYBP and TS-RING1B:BMI1-His, respectively (PreScission proteases were obtained from the Max-Planck Institute (MPI) of Biochemistry Protein Core facility). The untagged proteins were then dialyzed overnight against 25 mM Tris-HCl pH 7.5, 250 mM NaCl, 10% glycerol. RYBP was subjected to reverse Ni-NTA affinity chromatography and then the resulting flow through was mixed with RING1B:BMI1 at a 1:1 molar ratio. After a cation exchange chromatography, a final size-exclusion chromatography (SEC) step in 25 mM Tris-HCl, pH 7, 150 mM NaCl, 10% glycerol, 2 mM dithiothreitol (DTT) was performed. The vPRC1 sample utilized for cryo-EM analysis underwent an additional SEC run to assess the integrity of the complex. RYBP$_{T31A/F32A}$ and RYBP$_{R47A/R53A/R56A}$ mutants were expressed and purified following the same procedure of the wild-type protein. The purity of all the proteins was checked on polyacrylamide gels.

NG–RYBP and NG–RYBP$_{R47A/R53A/R56A}$ mutant proteins were expressed in *E. coli* Rosetta (DE3) pLysS cells. Transformed cells were grown at 37 °C in 10×P TB to approximately an OD$_{600}$ of 1.0 and expression was then induced with 0.5 mM isopropyl β-D-1-thiogalactopyranoside overnight (18 °C). Collected cells were lysed by sonication and the protein was then purified as described above.

*D. melanogaster* histones were purchased from *The Histone Source* at Colorado State University. For histone octamers, equimolar amounts of histones H2A, H2B, H4 and H3 were mixed and assembled into octamers in high salt buffer containing 10 mM Tris-HCl pH 7.5, 2 M NaCl, 1 mM EDTA, 5 mM β-mercaptoethanol. Subsequent SEC was performed to separate octamers from H3/H4 tetramers or H2A/H2B dimers[21].

### Reconstitution of mono- and tetranucleosomes

DNA templates used for mono and tetranucleosomes assembly contained one or four copies of the 147-bp-long nucleosome-positioning sequence 601 (ref. 22), respectively. PCR amplification with appropriate primer pairs was used to generate the different DNA templates used in this study. For cryo-EM samples, mononucleosomes were assembled on a 215-bp 5′-biotinylated 601 DNA fragment (5′-biotin-atatctcgggcttatgtgatgggaccctatacgcggccgcc-601-gcatgtattgaacagcgactcgggatat-3′). Mononucleosomes used in EMSA were reconstituted using a 288-bp long 5′-biotinylated 3′-ATTO647N-labeled DNA fragment that included a unique *Eco*RV restriction site 87 bp downstream of the 5′ biotin; following restriction enzyme

cleavage, a mononucleosome with the following DNA sequence was obtained (5′-atctcgggcttatgtgatggaccctatacgcggccgcc-601-gcatgtattgaacagcgactc-ATTO647N-3′)[23].

Tetranucleosomes were assembled on a 919-bp DNA fragment containing an array of four 200-bp repeats containing the 601 sequence separated by 53 bp linker DNA (5′-gtaaaacgacggcc agtgcca-agcttgcatgcctgcaggtcgactctagaggatccccgatatctcgggttatgtgatgg-accctatacgcggccgcc-601 -gcatgtattgaacagcgactcgggttatgtgatgg-accctatacgcggccgcc-601-gcatgtattgaacagcgactcgggttatg tgatg-gacccctatacgcggccgcc-601-gcatgtattgaacagcgactcgggttatgtgatgg-accctatacg cggccgcc-601-gcatgtattgaacagcgactcgggatatcgggtaccg-agctcgaattcgtaatcatggtcatagctgtttcctg-3′).

PCR products were purified on a MonoQ column (GE Healthcare), precipitated with ethanol and dissolved in the same high salt buffer used for octamers. Optimized ratios of octamer to DNA were mixed and nucleosomes were reconstituted by gradient and stepwise dialysis against low salt buffers to a final buffer containing 10 mM Tris-HCl, pH 7.5, 30 mM NaCl, 2 mM DTT. Reconstituted nucleosomes were analyzed on native agarose gels.

### H2A monoubiquitination in vitro

Ubiquitination assay was performed as described previously[13]. Briefly, 350 nM of nucleosomes were incubated in reactions containing E3, UBE1 (35 nM), UBCH5C (250 nM), ubiquitin (19 μM) and ATP (5 mM) in ubiquitination buffer (UB) containing 50 mM Tris-HCl, pH 7.5, 10 mM MgCl$_2$, 1 mM ZnCl$_2$ and DTT (0.5 mM). Reactions were incubated at 30 °C for 1.5 h or as indicated and were terminated by transfer of the sample to 4 °C. H2Aub1 formation was monitored by separation of reaction products on 16% polyacrylamide gels and visualizing by Coomassie staining. vPRC1 complex (168 nM) was used as E3 to generate H2Aub1-modified mononucleosomes used in cryo-EM analyses, whereas a recombinant minimal RING1B$_{1–130}$:BMI1$_{1–109}$ heterodimer (168 nM)[13] was used to generate H2Aub1 mononucleosomes for EMSA assays. To assess the contribution of RYBP to vPRC1 activity, we tested the E3 ligases RING1B:BMI1 full-length and vPRC1 wild-type and mutated complexes at a lower concentration of 35 nM. The assays with oligonucleosomes utilized an 88 nM tetranucleosome concentration, ensuring an equivalent substrate H2A concentration as in the mononucleosome assays. For quantification, ubiquitination reactions were performed in triplicate or more, and subjected to densitometric analysis. The H2Aub1 signal in each lane was background-corrected using Image Lab software (v.6.1) and normalized with respect to the H4 band. The relative amounts of H2Aub1 for all the different lanes were calculated with respect to the lane containing the highest amount (that is, 100%) of H2Aub1. Graphical representations were made with Prism v.9. Human UBE1, UBCH5C and ubiquitin were purchased from Boston Biochem.

### Cryo-EM sample preparation and data collection

To prevent denaturation or disassembly of particles at the air/water interface during grid preparation, we used Quantifoil R2/2 gold grids coated with at streptavidin crystal monolayer, prepared as described[24].

Samples for cryo-EM were prepared by mixing solutions of 350 μM Nuc with 3.5 mM of vPRC1 (Nuc/vPRC1 ratio of 1:10), or 350 μM Nuc$_{H2Aub1}$ with 1.75 mM vPRC1 (Nuc$_{H2Aub1}$/vPRC1 ratio of 1:5) in UB buffer and binding reactions were incubated for 1 h at 4 °C. Sample (5 μl) was then applied to an affinity grid and incubated in a closed Petri dish for 4 min to allow binding of the biotinylated nucleosomes to the streptavidin monolayer. Unbound sample was blotted away and 3 μl of 25 mM Tris/HCl pH 7.5, 40 mM KCl, 1 mM MgCl$_2$, 1 mM TCEP, 4% threalose, 0.04% B-OG, 0.01% NP40 (strep wash buffer) was added to the grid before mounting it in the humidity chamber of a Mark IV vitrobot (FEI) set to 5 °C and 95% relative humidity. In case of the vPRC1:Nuc$_{H2Aub1}$ sample, two additional washing steps were performed to remove all unbound protein components from the ubiquitination reaction

(that is E1, E2 and Ub). The first wash was performed with 4 μl UB buffer (10 s incubation) while the second was done in 4 μl Strep wash buffer without incubation. Samples were blotted from the streptavidin-coated surface for 3.5 s at a blot force of four, then plunge-frozen in liquid ethane.

Cryo-EM data were collected on an FEI Titan Krios microscope operated at 300 kV and equipped with a Gatan K3 direct electron detector operated in counting mode. Automated data collection was done using SerialEM[25]. For the vPRC1:Nuc sample, a total of 14,850 videos were collected at a nominal magnification of 105,000 and a pixel size of 0.8512 Å using a total exposure of 58.6 e⁻/Å² distributed over 38 frames and a target defocus range from −0.5 to −3.0 μm. For the vPRC1:Nuc$_{H2Aub1}$ sample, a total of 10,653 videos were collected at a nominal magnification of 81,000 and a pixel size of 1.094 Å using a total exposure of 54.0 e⁻/Å² distributed over 35 frames and a target defocus range from −0.5 to −3.0 μm.

### Cryo-EM data processing

Movies were aligned and corrected for beam-induced motion as well as dosage compensated using MotionCor2 (ref. [26]). Removal of the 2D streptavidin lattice was accomplished using a procedure in MATLAB that performs digital Fourier filtering on the motion-corrected, summed micrographs as described in ref. [27]. Cryo-EM data for vPRC1:Nuc were further processed with CryoSPARC (v.3.3.2)[28] (Extended Data Fig. 1a). In brief, micrographs were imported into CryoSPARC, where a contrast transfer function (CTF) estimation was performed. Based on maximum estimated resolution and CTF fit, the best 14,815 images were retained for the next steps. Considering that streptavidin grids have a comparatively low particle contrast, we applied a low threshold to pick as many particles as possible. First, 2,494,574 particles were picked from a subset of micrographs using blob particle picking, extracted with a box size of 256 × 256 and 2D classified to generate reference templates. Using all micrographs, a total of 6,723,019 particles were picked with template-based particle picking. Several rounds of 2D classification were applied to remove background and low-resolution particles (resolution cut-off 7 Å), resulting in a set of 147,764 clean particles (~2% of initial particles) showing density for RING1B:BMI1 heterodimer bound to mononucleosomes (Extended Data Fig. 1a). Two ab initio models were calculated and one of them (49% of the input particles) was subjected to heterogenous refinement (Extended Data Fig. 1a). One of the models (6.11 Å) was used as input for homogeneous refinement against all 147,764 particles (Extended Data Fig. 1a). One round of 3D classification into five classes was performed, but all classes looked similar (no distinct conformers were found) and one was used as starting volume for homogeneous refinement, local CTF refinement and nonuniform refinement, resulting in a 2.97 Å map. All final particles were reextracted with a box size of 280 × 280 pixels and refined (homogeneous and nonuniform refinement) against the previous volume, giving a final map at 2.91 Å resolution as determined from the gold-standard FSC criterion of 0.143 (ref. [29]; Extended Data Fig. 1b,c). This final map was postprocessed with a B-factor of −60 Å² and with DeepEMhancer[30] for map representations in figures. Additionally, we employed density modification and anisotropic B-factor sharpening techniques from Phenix[31] to generate enhanced maps that were subsequently utilized for comparison and model fitting, building and refinement processes.

Cryo-EM data for vPRC1:Nuc$_{H2Aub1}$ were processed with RELION v.3.1.1 (ref. [32]) (Extended Data Fig. 3a) where CTF estimation was performed (CTFFind4 (ref. [33])) and 10,653 images were selected based on their CTF parameters; 100 micrographs were used for blob-based auto-picking and, after a few rounds of 2D classification, clean classes were selected as templates for subsequent auto-picking on all micrographs. As explained above, we applied a low threshold to pick as many particles as possible. A total of 12,150,042 particles were picked and extracted using a box size of 224 × 224. After several rounds of 2D

classification, a cleaned set of 640,334 particles (~5% of initial particles) was obtained (Extended Data Fig. 3a). Of this particle set, 6,000 were used for initial 3D reconstruction and the map obtained was used as a starting model for a 3D classification, using all 640,334 particles. After removing classes and particles showing empty nucleosomes or low-resolution artefacts, a 3D auto-refine job was performed. We then refined this last map again against all 640334 particles from the 2D classification which resulted in a postprocessed volume at 3.18 Å resolution (Extended Data Fig. 3a). To improve the local resolution of the RYBP zinc finger domain and ubiquitin, we performed the same final 3D Auto-refine using a focused mask, covering the nucleosome and the better defined vPRC1 density on one surface. The resulting map after postprocessing had the same overall resolution of 3.18 Å (Extended Data Fig. 3b,c) but slightly better resolved features for RYBP and ubiquitin. This map was postprocessed applying a B-factor of −60 Å². As above, DeepEMhancer improved maps were used for making figures and for model fitting, building and refinement processes, we relied on the improved density-modified and sharpened maps produced with Phenix.

### Model building and map validation

For both the samples, available crystal structures were fitted into the experimental refined final maps using rigid-body fitting in UCSF ChimeraX v.1.4 (ref. [34]). The models were manually adjusted in Coot[35] and real-space refined in Phenix[36].

To build an initial model of vPRC1:Nuc, first the crystal structure of the *D. melanogaster* nucleosome core particle (NCP) (PDB: 6PWE)[37] was fitted into the density of vPRC1:Nuc using ChimeraX and was then rigid-body fitted in Phenix. The remaining empty density was then fitted with the RING1B:BMI1 heterodimer from the crystal structure of vPRC1 ubiquitylation module bound to NCP (PDB: 4R8P)[5]. This model was subjected to iterative rounds of real-space refinement in Phenix and some manual building and correction using COOT. Where cryo-EM density allowed it, the DNA double-strand was extended. The final refinement was performed in Phenix using the sharpened and density-modified map. Model building progress was monitored using map-to-model correlation coefficients, geometry indicators and the map-versus-model FSC (Table 1).

In the case of vPRC1:Nuc$_{H2Aub1}$, we first fitted the nucleosome core (PDB: 6PWE) into the map using ChimeraX. For ubiquitin and RYBP, the available NMR structure of ubiquitin bound to the related NZF domain (PDB: 1Q5W)[16] was used and fitted into the corresponding map region. Using COOT, the sequence of NZF was substituted with that of the RYBP zinc finger domain while the missing parts (residues 47–58) were built de novo and then fitted into the density. The model was subjected to iterative rounds of real-space refinement in Phenix and manual correction with COOT using density-modified and B-factor sharpened maps. Model building progress was monitored using map-to-model correlation coefficients, geometry indicators and the map-versus-model FSC (Table 1). Structures were visualized with UCSF ChimeraX.

### Purification of unmodified and H2Aub1 mononucleosomes for EMSA

Nuc and Nuc$_{H2Aub1}$ were assembled using a 288-bp-long 5′-biotinylated 3′-ATTO647N-labeled 601 DNA fragment as described above. These nucleosomes were used as substrates in H2A ubiquitination reactions (Nuc$_{H2Aub1}$) or mock reactions (Nuc) and were then purified using streptavidin magnetic beads (Dynabeads M-280 Streptavidin, Invitrogen). Briefly, beads were added to the reaction, incubated for 1 h at room temperature followed by removal of the supernatant and washes of the bead-bound nucleosomes with UB buffer (four washes of 2 min each)[23]. 3′-ATTO647N-labeled Nuc or Nuc$_{H2Aub1}$ mononucleosomes (201 bp) were then released from beads by cleavage with *Eco*RV restriction enzyme at 37 °C for 1 h. Nucleosome purity

and ubiquitination of H2A histone were checked by analysis on 16% polyacrylamide gels.

## Electrophoretic mobility shift assay

Protein binding affinity to mononucleosomes was measured using EMSA[20,38]. Increasing concentrations of protein (in the nanomolar range) were added to 0.2 nM ATTO647N-labeled unmodified or H2Aub1-modified mononucleosomes in 25 mM Tris-HCl pH 7.5, 50 mM NaCl, 0.05% Tween 20, 5 mM $MgCl_2$ and 4% glycerol. After 10 mins of incubation on ice, the samples were loaded on a 1% 0.4× TBE agarose gel, and gel electrophoresis was performed for 45 mins in 0.4× TBE buffer at 60 V. The gel was scanned with a Typhoon FLA 9500 using the Cy5 filter. Each EMSA experiment was performed in triplicate or more (the number of replicants is indicated in the figure legends). ATTO647N signal was used for densitometric analysis performed using the ImageJ v.12.24.12 software. Background correction and calculation of the fraction of bound nucleosomes was performed in MATLAB. The gel analysis involved delineating two distinct regions in each lane: (1) unbound nucleosomes (designated as 'unbound' boxes) and (2) shifted nucleosomes (referred to as 'bound,' encompassing everything above the 'unbound' region). Background correction for the unbound fractions involved subtracting the unbound fraction of the last lane of the gel from the signal of each bound fraction. The last lane was selected as a control, given that all nucleosomes in it were completely shifted. Similarly, for the bound fraction, the background correction was performed by subtracting to the signal of each bound fraction the one of the first lane where all the nucleosomes are unbound. To calculate the fraction of bound versus unbound nucleosomes, the numerical value corresponding to 'bound' nucleosomes in each lane was divided by the total signal (the sum of bound and unbound signals) within the same lane. To determine the apparent $K_d$, Hill function fitting was performed with Prism v.9 software.

## Reporting summary

Further information on research design is available in the Nature Portfolio Reporting Summary linked to this article.

## Data availability

The protein structure data reported in this study have been deposited in PDB under the accession codes PDB ID 8PP6 and 8PP7 and in the EMDB under accession codes EMD 17796 and EMD 17797. Source data are provided with this paper.

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

## Acknowledgements

We thank D. Bollschweiler and T. Schäfer of the Max-Planck Institute of Biochemistry Cryo-EM Facility for excellent technical support, S. Poepsel for sharing the protocol for streptavidin grid preparation, B.-G. Han and J. Basquin for advice and C. Long and other laboratory members for critical comments on the manuscript. This work was supported by the Deutsche Forschungsgemeinschaft (SFB1064, Project-ID 213249687) and the Max-Planck-Gesellschaft. The funders had no role in study design, data collection and analysis, decision to publish or preparation of the manuscript.

## Author contributions

M.C., E.K. and J.M. conceived the project. M.C. and C.B. designed, performed and analyzed the structural work. M.C. performed all biochemical assays. S.S. helped with protein expression. M.C., C.B. and J.M. wrote the manuscript.

## Funding

## Competing interests

The authors declare no competing interests.

## Additional information

**Extended data** is available for this paper at

**Supplementary information** The online version contains supplementary
material available at https://doi.org/10.1038/s41594-024-01258-x.

**Correspondence and requests for materials** should be addressed to
Christian Benda or Jürg Müller.

**Peer review information** *Nature Structural & Molecular Biology*
thanks the anonymous reviewers for their contribution to the
peer review of this work. Peer reviewer reports are available.
Dimitris Typas was the primary editor on this article and managed its
editorial process and peer review in collaboration with the rest of
the editorial team.

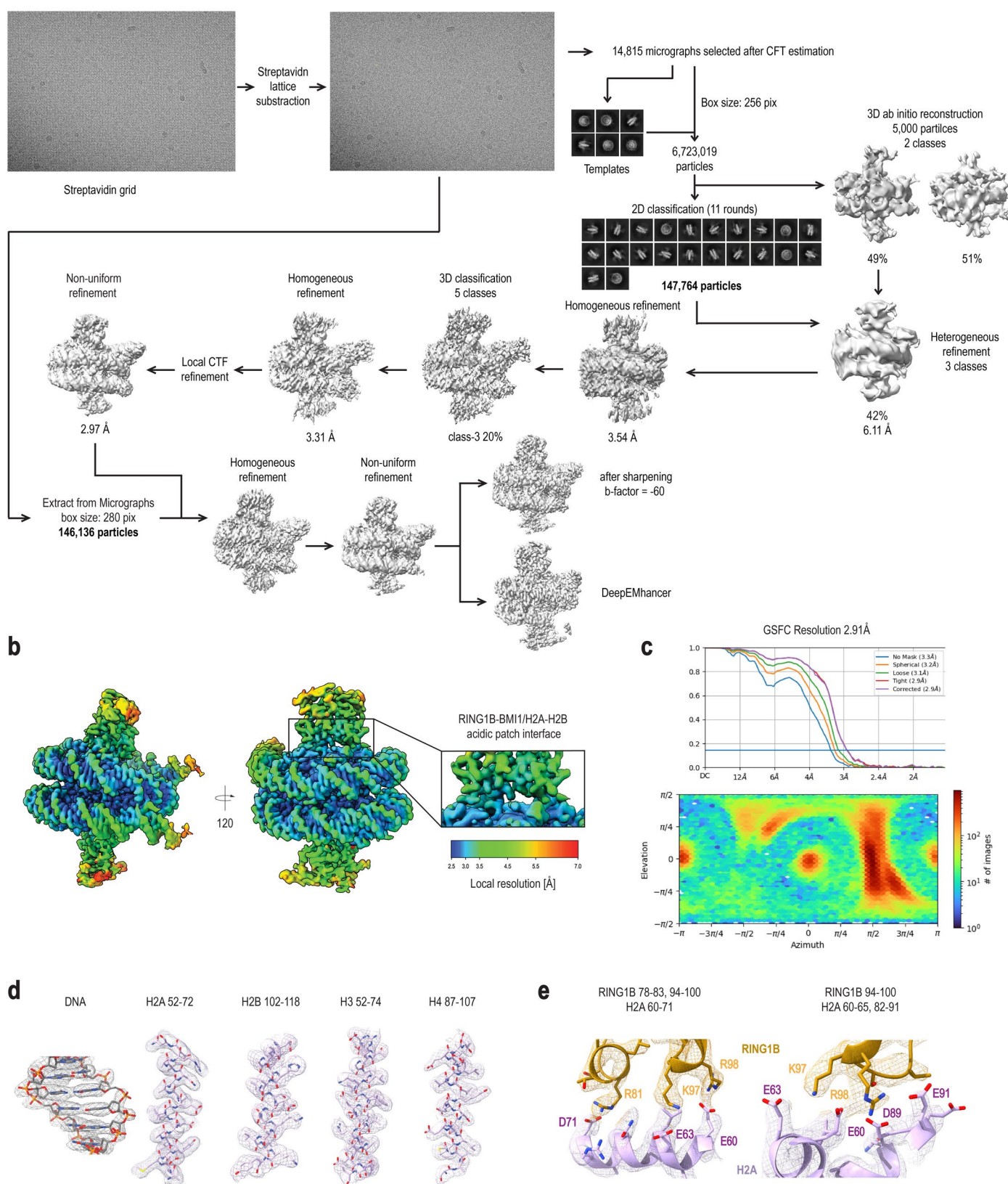

**Extended Data Fig. 1 | See next page for caption.**

**Extended Data Fig. 1 | CryoEM analysis of vPRC1:Nuc. (a)** CryoEM processing scheme, also described in Methods. (**b**) Local resolution estimation of the final deepEMhancer map is shown in two different orientations, related by a 120˚ rotation around a vertical axis. A zoom-in on the RING1B-BMI1/H2A-H2B acidic patch interference is displayed, with colors ranging from blue (high resolution) to red (low resolution). (**c**) Corresponding gold-standard Fourier shell correlation (GSFSC), as well as angular distribution plot obtained from CryoSPARC (color indicative of the number of particles, increasing from blue to red, in a defined orientation). (**d**) Representative regions of the DeepEMhancer density map for the nucleosome components (histones and DNA). DNA is in gray while histones are in pink. (**e**) Representative regions of the DeepEMhancer density map for the interfaces involving RING1B interaction with the H2A acidic patch. H2A histone in pink while RING1B in gold.

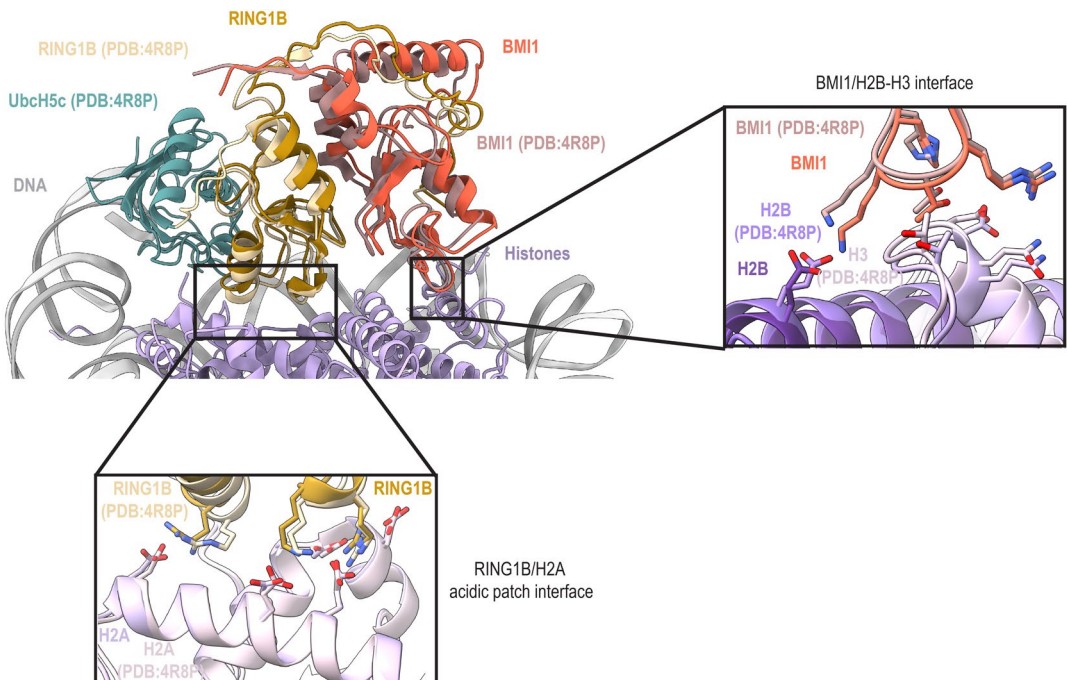

**Extended Data Fig. 2 | Comparison of the structures of vPRC1 and RING1B-UBCH5C:BMI1 bound to an unmodified nucleosome.** Top left, superposition of the vPRC1:Nuc complex cryo-EM structure in this study and the RING1B-UBCH5C:BMI1:Nuc complex structure (PDB: 4R8P) previously determined by X-ray crystallography[5]. The two structures were superimposed via H2A but for clarity in the overview only the nucleosome model determined by cryo-EM is shown. Below, zoom-in view of the RING1B/H2A acidic patch interface; right, zoom-in view of the BIM1/H2B-H3 interface. The vPRC1 cryoEM structure follows the same color code as depicted in Fig. 1 and the PDB: 4R8P structure is shown in lighter shades of the same colors. Note that McGinty et al used a RING1B-UBCH5C fusion protein in complex with BMI1 to stabilize the complex on the nucleosome; UBCH5C is shown in teal. The superposition illustrates that the RING1B:BMI1 ring finger heterodimer in the two structures engages with the nucleosome acidic patch with a highly similar binding geometry and through the identical amino acid contacts.

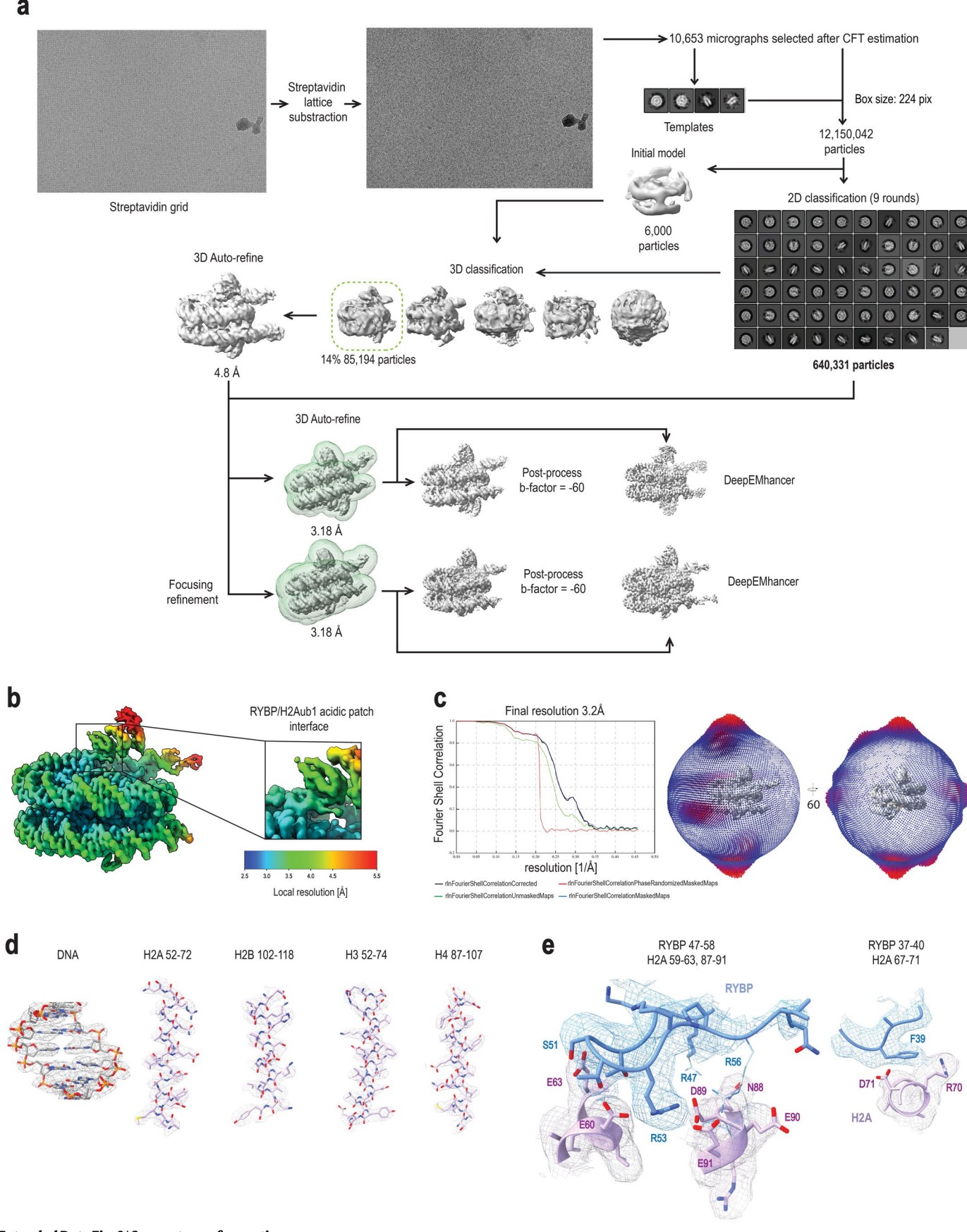

**Extended Data Fig. 3 | See next page for caption.**

**Extended Data Fig. 3 | CryoEM analysis of vPRC1:Nuc$_{H2Aub1}$.** (**a**) Processing scheme, also described in Methods. Two final reconstructions were obtained in this study: overall vPRC1:Nuc$_{H2Aub1}$ map and one side vPRC1:Nuc$_{H2Aub1}$ map from focusing refinement on one surface of the nucleosome disc (mask indicated in green). The two refined maps were further post-processed with a b-factor of −60 Å$^2$ and with DeepEMhancer. (**b**) Local resolution estimation of the final one side density DeepEMhancer map; RYBP/H2A acidic patch interference is highlighted showing a ~ 3.5 Å resolution with colors ranging from blue (high resolution) to red (low resolution). (**c**) Corresponding gold-standard Fourier shell correlation (GSFSC) curves, as well as spherical distribution plot obtained from CryoSPARC (each bar has a height and color indicative of the number of particles (increasing from blue to red) in a defined orientation. (**d**) Representative regions of the DeepEMhancer density map for the nucleosome components (histones and DNA). DNA is in gray while histones are in pink. (**e**) Representative regions of the DeepEMhancer density map for the interfaces involving RYBP interactions with the H2A acidic patch. H2A is in pink while RYBP in blue.

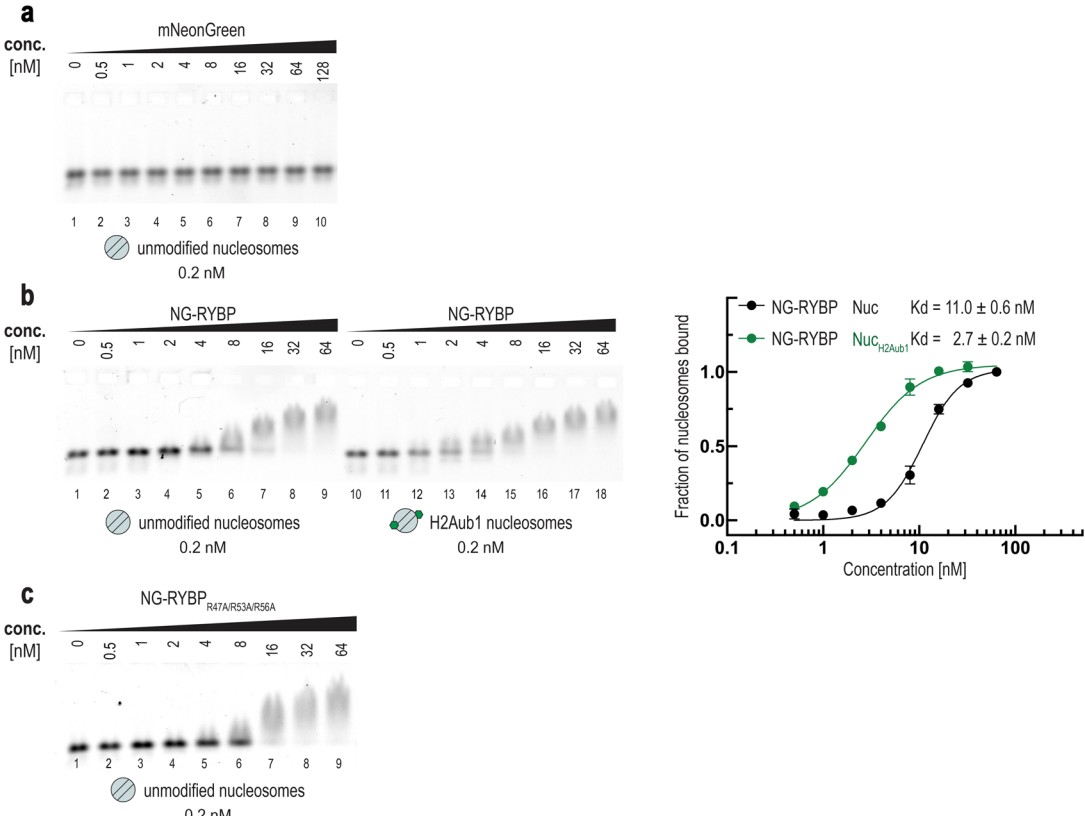

**Extended Data Fig. 4 | Binding of mNeonGreen, NG–RYBP or NG-RYBPR47A/ R53A/R56A proteins to unmodified or H2Aub1-modified mononucleosomes.**
(**a**) mNeonGreen (NG) protein lacks nucleosome-binding activity. Binding reactions with the indicated concentrations of NG and 0.2 nM 647N-ATTO-labeled unmodified mononucleosomes, analyzed by EMSA on 1.0% agarose gels. (**b**) NG–RYBP shows higher binding affinity for H2Aub1-modified nucleosomes than for unmodified nucleosomes. Left, binding assays by EMSA as in (**a**). Binding of NG–RYBP to H2Aub1-modified mononucleosomes as in Fig. 2a,b, is

compared to the binding of the same wild-type NG–RYBP protein to unmodified mononucleosomes. Right, quantitative analysis of EMSA data by densitometry of 647N-ATTO signal from independent experiments (n = 3); error bars, SEM. The residual binding by the NG–RYBP protein on unmodified mononucleosomes in lanes 7–9 appears to be independent of RYBP-H2A acidic patch interactions because at high concentrations binding is also observed with NG–RYBP$_{R47A/R53A/R56A}$ on unmodified mononucleosomes (**c**); the nature of this RYBP interaction with nucleosomes is currently unknown.

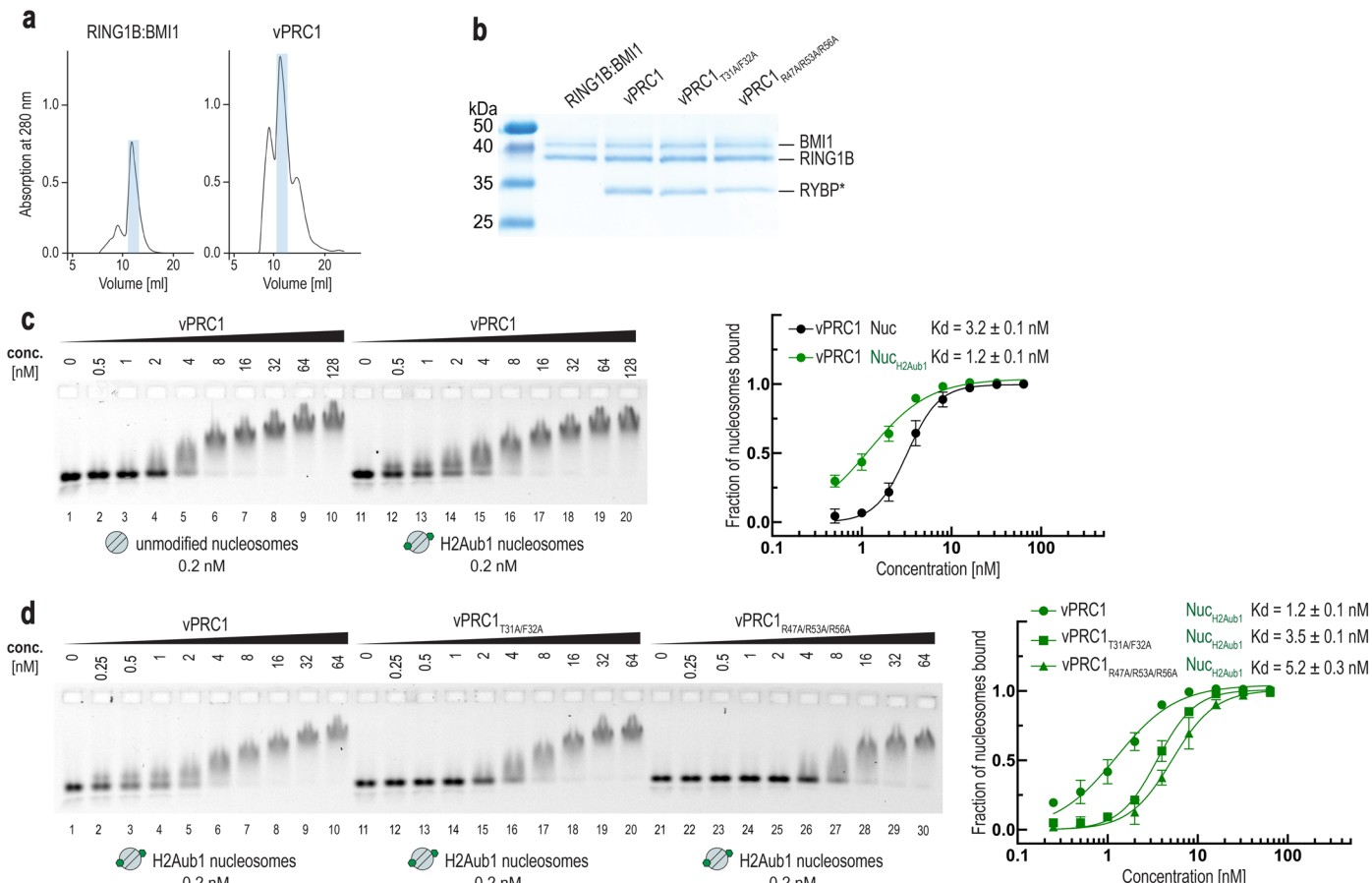

**Extended Data Fig. 5 | High-affinity binding of vPRC1 to H2Aub1-modified nucleosomes relies on RYBP–H2Aub1 contacts. (a)** Size-exclusion chromatography profiles of reconstituted RING1B:BMI1 full-length and vPRC1 wild-type complex (S200 column, absorption at 280 nm). Fractions used in all binding and activity assays are marked in blue. Note that the vPRC1 sample utilized for Cryo-EM analysis underwent an additional SEC run to assess the integrity of the complex (see Fig. 1b). **(b)** Coomassie-stained SDS gel showing the RING1B:BMI1 full-length dimer, wild-type and mutant vPRC1 complex samples used in biochemical assays. RYBP* denotes the wild-type or mutant form of RYBP, as indicated in the complex name above each lane. **(c)** vPRC1 shows higher binding affinity for H2Aub1-modified nucleosomes than for unmodified nucleosomes. Left, binding reactions with the indicated concentrations of vPRC1 and 0.2 nM 647N-ATTO-labeled unmodified or H2Aub1-modified

mononucleosomes, analyzed by EMSA on 1.0% agarose gels. Right, quantitative analysis of EMSA data by densitometry of 647N-ATTO signal from independent experiments (n = 3); error bars, SEM. **(d)** RYBP interactions with ubiquitin and the H2A acidic patch account for the high binding affinity of vPRC1 complex on H2Aub1-modified nucleosomes. Left, binding assays by EMSA as in **(c)**. Binding of wild-type vPRC1, vPRC1$_{T31A/F32A}$ and vPRC1$_{R47A/R53A/R56A}$ to H2Aub1-modified mononucleosomes. Right, quantitative analysis of EMSA results as in **(c)** from independent experiments (n = 3); error bars, SEM. Note that the binding observed at higher concentrations of wild-type or mutant vPRC1 (lanes 8–10, 18–20 and 28–30) likely represents the sum of binding interactions due to RING1B:BIMI1 Ring-finger-acidic patch contacts (Fig. 2a) and RYBP-nucleosome contacts unrelated to the RYBP–H2Aub1 interaction characterized in this study (Extended Data Fig. 4c).

Christian Benda

# Reporting Summary

## Statistics

For all statistical analyses, confirm that the following items are present in the figure legend, table legend, main text, or Methods section.

| n/a | Confirmed | |
|---|---|---|
| ☐ | ☒ | The exact sample size (*n*) for each experimental group/condition, given as a discrete number and unit of measurement |
| ☐ | ☒ | A statement on whether measurements were taken from distinct samples or whether the same sample was measured repeatedly |
| ☒ | ☐ | The statistical test(s) used AND whether they are one- or two-sided *Only common tests should be described solely by name; describe more complex techniques in the Methods section.* |
| ☒ | ☐ | A description of all covariates tested |
| ☒ | ☐ | A description of any assumptions or corrections, such as tests of normality and adjustment for multiple comparisons |
| ☐ | ☒ | A full description of the statistical parameters including central tendency (e.g. means) or other basic estimates (e.g. regression coefficient) AND variation (e.g. standard deviation) or associated estimates of uncertainty (e.g. confidence intervals) |
| ☒ | ☐ | For null hypothesis testing, the test statistic (e.g. *F*, *t*, *r*) with confidence intervals, effect sizes, degrees of freedom and *P* value noted *Give P values as exact values whenever suitable.* |
| ☒ | ☐ | For Bayesian analysis, information on the choice of priors and Markov chain Monte Carlo settings |
| ☒ | ☐ | For hierarchical and complex designs, identification of the appropriate level for tests and full reporting of outcomes |
| ☒ | ☐ | Estimates of effect sizes (e.g. Cohen's *d*, Pearson's *r*), indicating how they were calculated |

*Our web collection on statistics for biologists contains articles on many of the points above.*

## Software and code

Policy information about availability of computer code

| Data collection | SerialEM 4.0 |
|---|---|
| Data analysis | RELION 3.1<br>MotionCorr2 1.1<br>CTFFIND 4.1.13.<br>Coot 0.9.8.7<br>PHENIX 1.20<br>UCSF ChimeraX 1.4.<br>ImageJ 12.24.12<br>Matlab R2022B<br>Prism9 9.5.1<br>DeepEMhancer<br>Image Lab v6.1 |

For manuscripts utilizing custom algorithms or software that are central to the research but not yet described in published literature, software must be made available to editors and reviewers. We strongly encourage code deposition in a community repository (e.g. GitHub). See the Nature Portfolio guidelines for submitting code & software for further information.

## Data

Policy information about availability of data

All manuscripts must include a data availability statement. This statement should provide the following information, where applicable:
- Accession codes, unique identifiers, or web links for publicly available datasets
- A description of any restrictions on data availability
- For clinical datasets or third party data, please ensure that the statement adheres to our policy

> The protein structure data reported in this study have been deposited in PDB under the accession codes PDB ID 8PP6 and 8PP7 and in the EMDB under accession codes EMD-17796 and EMD-17797.
> Initial models used for CryoEM model building (PDB code): 6PWE, 4R8P, 1Q5W

## Research involving human participants, their data, or biological material

Policy information about studies with human participants or human data. See also policy information about sex, gender (identity/presentation), and sexual orientation and race, ethnicity and racism.

| | |
|---|---|
| Reporting on sex and gender | N/A |
| Reporting on race, ethnicity, or other socially relevant groupings | N/A |
| Population characteristics | N/A |
| Recruitment | N/A |
| Ethics oversight | N/A |

Note that full information on the approval of the study protocol must also be provided in the manuscript.

# Field-specific reporting

Please select the one below that is the best fit for your research. If you are not sure, read the appropriate sections before making your selection.

☒ Life sciences ☐ Behavioural & social sciences ☐ Ecological, evolutionary & environmental sciences

For a reference copy of the document with all sections, see nature.com/documents/nr-reporting-summary-flat.pdf

# Life sciences study design

All studies must disclose on these points even when the disclosure is negative.

| | |
|---|---|
| Sample size | The numbers of particles used for cryo-EM reconstructions are indicated in Extended Data Table 1<br>For all EMSAs and enzyme activity assays, we conducted experiments with three or more replicates to ensure reproducibility.<br>The consistent results across all replicates suggests that the number of replicates was sufficient in all experiments.<br>No statistical methods were used to predetermine the sample size. |
| Data exclusions | No data were excluded |
| Replication | For quantitative measurements, experiments were performed at least in triplicate. All of them were successful. |
| Randomization | not relevant, no grouped samples |
| Blinding | not relevant, no grouped samples |

# Reporting for specific materials, systems and methods

We require information from authors about some types of materials, experimental systems and methods used in many studies. Here, indicate whether each material, system or method listed is relevant to your study. If you are not sure if a list item applies to your research, read the appropriate section before selecting a response.

## Materials & experimental systems

| n/a | Involved in the study |
|-----|----------------------|
| ☒ | ☐ Antibodies |
| ☐ | ☒ Eukaryotic cell lines |
| ☒ | ☐ Palaeontology and archaeology |
| ☒ | ☐ Animals and other organisms |
| ☒ | ☐ Clinical data |
| ☒ | ☐ Dual use research of concern |
| ☒ | ☐ Plants |

## Methods

| n/a | Involved in the study |
|-----|----------------------|
| ☒ | ☐ ChIP-seq |
| ☒ | ☐ Flow cytometry |
| ☒ | ☐ MRI-based neuroimaging |

## Eukaryotic cell lines

Policy information about cell lines and Sex and Gender in Research

| | |
|---|---|
| Cell line source(s) | Cell line (Trichoplusia ni)<br>HighFive cell line for expression Invitrogen Product nr.:<br>B85502<br>BTI-Tn-5B1-4<br>(RRID:CVCL_C190)<br>Protein expression<br>Cell line (Spodoptera frugiperda)<br>Sf21 cell line for<br>Baculovirus production<br>Invitrogen Product nr.:<br>1149701<br>(RRID:CVCL_0518)<br>Baculovirus production for protein expression |
| Authentication | Commercial cell lines. Authentication is performed by the manufacturer |
| Mycoplasma contamination | The cell line were not tested for mycoplasma contamination |
| Commonly misidentified lines (See ICLAC register) | No commonly misidentified lines were used |

## Plants

| | |
|---|---|
| Seed stocks | *Report on the source of all seed stocks or other plant material used. If applicable, state the seed stock centre and catalogue number. If plant specimens were collected from the field, describe the collection location, date and sampling procedures.* |
| Novel plant genotypes | *Describe the methods by which all novel plant genotypes were produced. This includes those generated by transgenic approaches, gene editing, chemical/radiation-based mutagenesis and hybridization. For transgenic lines, describe the transformation method, the number of independent lines analyzed and the generation upon which experiments were performed. For gene-edited lines, describe the editor used, the endogenous sequence targeted for editing, the targeting guide RNA sequence (if applicable) and how the editor was applied.* |
| Authentication | *Describe any authentication procedures for each seed stock used or novel genotype generated. Describe any experiments used to assess the effect of a mutation and, where applicable, how potential secondary effects (e.g. second site T-DNA insertions, mosiacism, off-target gene editing) were examined.* |

