## [Peer Review File · Nature Structural & Molecular Biology]

Peer Review Information

Manuscript Title: Structural basis of the histone ubiquitination read-write mechanism of RYBP-PRC1

Corresponding author name(s): Juerg Mueller, Christian Benda

Reviewer Comments & Decisions:

Decision Letter, initial version:

Message: 21st Aug 2023

Dear Dr Mueller,

Thank you again for submitting your manuscript "Structural basis of the histone ubiquitination read-write mechanism of RYBP-PRC1". I apologise for the delay in responding, which resulted from the difficulty in obtaining suitable referee reports. Nevertheless, we now have comments (below) from the 3 reviewers who evaluated your paper. In light of these reports, we remain interested in your study and would like to see your response to the comments of the referees, in the form of a revised manuscript.

You will see that reviewers while all reviewers find the potential mechanism interesting, there are several concerns and suggestions that should be addressed in a revised manuscript. More specifically, both reviewer #2 and #3 request experimental validation in binding or activity assays of the residues predicted to regulate the interaction with ubiquitin with relevant mutants. Moreover, reviewer #3 raises the critical importance of validating the data in the relevant context of dinucleosomes (or longer polynucleosomes). While we think that producing structures in that context is out of scope for this study, we editorially agree that reproducing (some of) the binding data with unmodified/monoubiquitylated K119 dinucleosomes will be important for the successful peer-review of this manuscript. Finally, we ask you to please take heed of the presentation suggestions by reviewer #1 and perform the additional controls suggested by reviewer #2 (points 2 and 4).

Please be sure to address/respond to all concerns of the referees in full in a point-by-point response and highlight all changes in the revised manuscript text file.

We appreciate the requested revisions are extensive. We thus expect to see your revised manuscript within 3-6 months. If you cannot send it within this time, please let us know. We will be happy to consider your revision as long as nothing similar has been accepted for publication at NSMB or published elsewhere. Should your manuscript be substantially delayed without notifying us in advance and your article is eventually published, the received date would be that of the revised, not the original, version.

Reporting Summary:

When submitting the revised version of your manuscript, please pay close attention to our [href="https://www.nature.com/nature-portfolio/editorial-policies/image-integrity">Digital Image Integrity Guidelines. and to the following points below:](https://www.nature.com/nature-portfolio/editorial-policies/image-integrity)

While we encourage the use of color in preparing figures, please note that this will incur a charge to partially defray the cost of printing. Information about color charges can be

found at <http://www.nature.com/nsmb/authors/submit/index.html#costs>

We require deposition of coordinates (and, in the case of crystal structures, structure factors) into the Protein Data Bank with the designation of immediate release upon publication (HPUB). Electron microscopy-derived density maps and coordinate data must be deposited in EMDB and released upon publication. Deposition and immediate release of NMR chemical shift assignments are highly encouraged. Deposition of deep sequencing and microarray data is mandatory, and the datasets must be released prior to or upon publication. To avoid delays in publication, dataset accession numbers must be supplied with the final accepted manuscript and appropriate release dates must be indicated at the galley proof stage. Please find the complete NRG policies on data availability at <http://www.nature.com/authors/policies/availability.html>.

[Redacted]

Sincerely,

Dimitris Typas
Associate Editor
Nature Structural & Molecular Biology
ORCID: 0000-0002-8737-1319

Referee expertise:

Referee #1: biochemistry and structural biology of chromatin proteins

Referee #2: PRC1/2 (structural and biochemistry)

Referee #3: structural biology of nucleosome-bound chromatin complexes, including ubiquitin-binding and modifying proteins

Reviewers' Comments:

Reviewer #1:

Remarks to the Author:

The authors report the cryoEM structures of RYBP-PRC1 (vPRC1) bound to unmodified and H2AK119ub1-modified mono-nucleosomes. vPRC1 plays a critical role in Polycomb repression as it can both ubiquitinate H2AK119 and binds to the ubiquitinated H2AK119ub1 nucleosomes via its RYBP or YAF2 subunit. Structures of the ubiquitination module of the complex bound to an unmodified nucleosome have been previously reported. This manuscript provides the first structure of the RYPBP-PRC1 bound to an H2AK119ub1 nucleosome and discovers two distinct modes of binding for the complex. The complex binds to unmodified nucleosomes via RING1B (consistent with previous structures) and to H2AK119ub1-modified nucleosomes via RYBP, which interacts with both ubiquitin and the H2A acidic patch. The two modes of binding are mutually exclusive leading the authors to propose that RYBP-PRC1 binds to H2Aub1-modified nucleosomes via RYBP and to adjacent unmodified nucleosomes via RING1B, providing a structural basis for how RYBP-PRC1 read-write propagates H2Ak119ub1.

This is an excellent short paper that provides clear new insight into the mechanism of vPRC1 read-write whereby the binding of RYBP-PRC1 to K119ub1 nucleosomes leaves the RING1B/BMI1 module free to bind to the acidic patch of H2A on an adjacent nucleosome to modify H2AK119. The experiments are of very high technical quality and both the structural and biochemical data are convincing and support the authors conclusions. Some side chain densities for RYBP residues that interact with the acidic patch on the nucleosome lack clear density, but the authors' modeling for the binding of these side chains to the acidic patch is supported by their binding assays. I support publication in NSMB enthusiastically and only have minor points for the authors to consider.

-Supplementary figure 1. A separate panel showing the purification of each RING1B:BMI1 and vPRC1 should be included. This is important since the cryoEM structures are missing large segments of various subunits and would allow the reader to better judge the quality of the complexes used for making grids.

-A comparison of the RING1B interactions with the nucleosome acidic patch in the current study with previous structures would be useful (perhaps as a suppl figure).

Line 37. Typo. Ref 2 does not address PRC1 or its catalytic activity.

Reviewer #2:

Remarks to the Author:

The authors have solved the structure of RYBP-PRC1 in presence of nucleosomes either unmodified or ubiquitinated on H2A. The structure in presence of unmodified nucleosome is consistent with previous report based on crystallography but they did not detect any density for RYBP. In contrast, the structure in presence of H2Aub-Nuc reveals an additional density assigned to RYBP and ubiquitin. This enables the authors to define RYBP's interaction with H2A acidic patch and ubiquitin. Finally, they showed that RYBP has more affinity for nucleosomes, in particular H2Aub-Nuc, than the RING1B:BMI1 dimer.

This is a very interesting study with a clear message and well supported by the data. We nonetheless recommend a few additional controls for publication.

Specific comments

1- line 120/121/124/196/198... indicates that R46 is mutated, we assume that the authors meant R47 and not V46?

2- Figure 2a, one essential condition is missing: comparing the affinity of RING1B:BMI1 on modified and unmodified nucleosomes. Does the presence of ubiquitin disfavor the binding of RING1:BMI1, is it neutral, etc.

3- Figure 2, more important than the mutations that abolish the interaction with the acid patch, are the mutations which specifically affect the interaction with ubiquitin to determine whether they are also necessary for the interaction between RYBP and H2Aub1 (the experiments presented only suggest that these residues are not sufficient for this interaction).

4- EMSA should also be performed with RYBP-PRC1 (WT and mutants) and not only with RYBP alone as its binding properties might be different when part of the PRC1 complex.

Reviewer #3:

Remarks to the Author:
NSMB-BC47936

Previous studies had shown that monoubiquitination of histone H2A-K119 by the heterodimeric PRC1 E3 ligase (RING1B/BMI1) could be enhanced by RYBP, which associates with one H2A-ubiquitinated nucleosome and recruits PRC1 subunit RING1B to ubiquitinate an adjacent, unmodified nucleosome. This mechanism explains how H2A-K119 ubiquitination can "spread" along a region of chromatin. A previous structural study by Tan and colleagues had shown how RING1B/BMI1 bind to a nucleosome and target H2A-K119 for ubiquitination. The aim of the work described in this manuscript is to provide mechanistic insights into the mechanism by which RYBP associates with PRC1 and promotes ubiquitination of adjacent nucleosomes. The authors have determined cryo-EM structures from grids prepared with RYBP-PRC1 with unmodified and with H2A-ubiquitinated nucleosomes. The map obtained with unmodified nucleosomes shows RING1B/BMI1 contacting the nucleosome acidic patch in essentially the same orientation as seen in the prior crystal structure, with no density corresponding to RYBP. The map obtained with nucleosomes containing H2A-K119Ub show density for a fragment of RYBP alone, with no density corresponding to RING1B/BMI1. A short stretch of RYBP binds to the nucleosome acidic patch via arginine residues while the zinc finger domain binds ubiquitin. The authors propose a "read-write" mechanism in which ubiquitin drives binding of RYBP, which displaces PRC1 from the nucleosome acidic patch and directs PRC1 to monoubiquitinate H2A in neighboring unmodified nucleosomes. While this is an intriguing mechanism that would provide an elegant explanation for the role of RYBP in promoting spreading of H2A ubiquitination, there is insufficient evidence in this manuscript to support

the model. Specific points are as follows:

Major

1. A major issue is that all the structural and biochemical data were obtained with mononucleosomes, which is not the relevant substrate. The authors test the effects of mutations on binding to mononucleosomes, whereas their model is based on simultaneous binding of RYBP to one nucleosome and PRC1 to another. It seems likely that there are additional contacts with a dinucleosome that do not form with single nucleosomes, for example with linker DNA. It is too much of a stretch to extrapolate from binding affinity measurements on a single nucleosome to the affinity of the complex for a dinucleosome. In order to verify that the observed acidic patch contacts are indeed important, the experiments need to be done on a dinucleosome or longer polynucleosome template. Protocols are available for assembling a dinucleosome template in which one nucleosome contains H2A-Ub, which would be an appropriate substrate for testing the effects of mutations.
2. Another issue is the reliance on binding, rather than activity assays. Since the model makes a prediction about activity, the effect of mutating the basic residues in RYBP should be tested on a dinucleosome or polynucleosome template.
3. It seems that one prediction of the authors' model is that two vPRC1 complexes could bind to an unmodified dinucleosome, whereas just a single complex would bind to a dinucleosome (or multiply both by 2 if both faces are occupied). This could be a way to test the proposed competition model.
4. In structure of vPRC1:NucH2Aub1, RYBP F32, I25 and T31 were identified to be essential for binding to Ub. Are there experiments (by the authors or in the literature) that verify the importance of these residues for either nucleosome binding or enzymatic activity?
5. As described in the Methods section on EM data processing for vPRC1:Nuc and vPRC1:NucH2Aub1, only ~2% (147,764/6,723,019) of particles and ~5% (640,334/12,150,042) of particles, respectively, were retained after 2D classification. Some justification or explanation should be given for the very small percentage of particles used to generate the 3D reconstruction.
6. It is not clear why the structure of PRC1 bound to an unmodified nucleosome was reported given that there is already a higher resolution crystal structure of the complex. Some explanation should be given for including it.
7. It would be helpful to readers to provide a diagram of the model for the read-write mechanism.

Minor

1. The functional difference between the canonical and variant PRC1 complex should be described explicitly in the introduction.
2. In Figure 1g, 1e and 1h, protein names should be appropriately labeled for clearer presentation.
3. In Page 8, line 172, vPRC1:NclH2Aub1 should be vPRC1:NucH2Aub1.

4. In the supplementary data, page 3, line 33, 3.5 A should be 3.5 Å.
5. In page 17, line 372, the reference is not inserted properly.

Author Rebuttal to Initial comments

Point-by-point response

Reviewer #1

“This is an excellent short paper that provides clear new insight into the mechanism of vPRC1 read-write whereby the binding of RYBP-PRC1 to K119ub1 nucleosomes leaves the RING1B/BMI1 module free to bind to the acidic patch of H2A on an adjacent nucleosome to modify H2AK119. The experiments are of very high technical quality and both the structural and biochemical data are convincing and support the authors conclusions.”

Thank you for your kind words and support.

Minor point 1

Supplementary figure 1. A separate panel showing the purification of each RING1B:BMI1 and vPRC1 should be included. This is important since the cryoEM structures are missing large segments of various subunits and would allow the reader to better judge the quality of the complexes used for making grids.

Our response: We have now added gel filtration diagrams and Coomassie-stained gels for purified RING1B:BMI1 complex and wild-type vPRC1 or mutant vPRC1 complexes that were used for EMSA and ubiquitination assays. Data is shown side-by-side in **Extended Data Figure 5a, b**. This is in addition to figures in the original submission showing the gel filtration purification profile of vPRC1 (Fig. 1b) and a Coomassie-stained gel of vPRC1 assembled with unmodified and H2Aub1-modified mononucleosomes that were used for Cryo-EM grid preparation (Fig. 1c).

Minor point 2

A comparison of the RING1B interactions with the nucleosome acidic patch in the current study with previous structures would be useful (perhaps as a suppl figure).

Our response: In **Extended Data Figure 2**, we have added images showing the superposition of the two structures, i.e. that of RING1B:BMI1:RYBP bound to an unmodified nucleosome as determined by cryo-EM in our study and that of RING1B-UBCH5C:BMI1 bound to a nucleosome determined by X-ray crystallography in Song Tan’s lab (McGinty et al, 2014). The superposition illustrates that the RING1B:BMI1 Ring finger heterodimer in the two structures engages the nucleosome acidic patch with a highly similar binding orientation and through the identical amino acid contacts.

Minor point 3

Line 37. Typo. Ref 2 does not address PRC1 or its catalytic activity.

Our response: We apologize for this oversight. The text is updated to state the correct reference (Wang et al, 2004).

Reviewer #2

“This is a very interesting study with a clear message and well supported by the data.”

We appreciate your interest and support of our study.

Point 1

line 120/121/124/196/198... indicates that R47 is mutated, we assume that the authors meant R47 and not V46?

Our response: We thank the reviewer for spotting this typo. Like in the figure, it should have been R47A in all instances in the text, and we have now corrected this.

Point 2

Figure 2a, one essential condition is missing: comparing the affinity of RING1B:BMI1 on modified and unmodified nucleosomes. Does the presence of Ubiquitin disfavor the binding of RING1:BMI1, is it neutral, etc.

Our response: We agree with the Reviewer and had in fact performed this control in parallel on the same gel that we showed in Figure 2a. However, for simplicity, we did not include the data in our original submission. In the revised **Figure 2a**, we now show the full gel image that includes this control. Indeed, the presence of Ubiquitin appears to be ‘neutral’ and does not seem to interfere with RING1B:BMI1-mediated binding to the nucleosome. This reinforces the interpretation that the high binding affinity created by the combined interactions of the RYBP zinc finger with Ubiquitin and of the RYBP[47-58] loop with the nucleosome acidic patch is the main determinant that directs vPRC1 to bind H2Aub1-modified mononucleosomes via RYBP.

We now discuss this in the revised text as follows: “Of note, H2Aub1 does not appear to impact on RING1B:BMI1 binding to nucleosomes; the RING1B:BMI1 dimer bound H2Aub1-modified and unmodified nucleosomes with very similar affinity (**Figure 2a**, compare lanes 11-20 with lanes 21-30).”

Point 3

Figure 2, more important than the mutations that abolish the interaction with the acid patch, are the mutations which specifically affect the interaction with Ubiquitin to determine whether they are also necessary for the interaction between RYBP and H2Aub1 (the experiments presented only suggest that these residues are not sufficient for this interaction).

Our response: This point was also raised by reviewer #3. To address this, we have analyzed the binding of the RYBP[T31A/F32A] mutant protein to H2Aub1 mononucleosomes. RYBP residues T31 and F32 were previously shown to be critical for the interaction of RYBP with free Ubiquitin (Arrigoni et al, 2006) and with H2Aub1 nucleosomes (Zhao et al, 2020). Specifically, the T31A/F32A double mutation in RYBP was previously reported to reduce RYBP binding to H2Aub1 nucleosomes in pull-down experiments (Zhao et al, 2020) and to impede monoubiquitination of unmodified H2A in nucleosomal arrays that contained pre-installed H2Aub1 on a subset of nucleosomes (Zhao et al, 2020). We performed additional EMSA experiments that show that the T31A/F32A double mutation in RYBP reduces its binding affinity to H2Aub1 mononucleosomes by about 3-fold. These data are presented in **Figure 2b** (lanes 11-19) (please also see response to Point 4 below).

Point 4

EMSA should also be performed with RYBP-PRC1 (WT and mutants) and not only with RYBP alone as its binding properties might be different when part of the PRC1 complex.

Our response: Thank you for the suggestion. We have now performed EMSAs to compare the binding of wild-type vPRC1 and mutant vPRC1 on H2Aub1-modified mononucleosomes. These experiments are presented in **Extended Data Figure 5d** in the revised manuscript. In brief, we found that vPRC1 containing RYBP in which we mutated either the acidic patch-contacting residues (RYBP_{R47A/R53A/R56A}) or the Ubiquitin-contacting residues (RYBP_{T31A/F32A}) both showed a reduced binding affinity compared to wild-type vPRC1. The binding affinity of vPRC1 with these mutated RYBP proteins on H2Aub1 mononucleosomes is comparable to that of vPRC1 on unmodified mononucleosomes that we also analysed (**Extended Data Figure 5c**).

We further extended these analyses by also performing ubiquitination assays with vPRC1 on short tetra-nucleosome arrays where we monitored the kinetics of H2Aub1 formation. We rationalized that, in the array, upon monoubiquitination of H2A on a first nucleosome, the proposed read-write mechanism should facilitate monoubiquitination of linked nucleosomes in an RYBP-dependent manner. These experiments on arrays showed that wild-type vPRC1 exerted greatly enhanced activity for H2Aub1 formation compared to the RING1B:BMI1 dimer alone. Importantly, the activity of vPRC1 containing RYBP_{R47A/R53A/R56A} was strongly reduced (the activity of vPRC1 containing RYBP_{T31A/F32A} was reduced only during the early phases of the time course experiment) (**Figure 2c**). This directly demonstrates the importance of the RYBP arginine loop in promoting H2A monoubiquitination on oligonucleosome templates.

Reviewer #3

Major point 1

A major issue is that all the structural and biochemical data were obtained with mononucleosomes, which is not the relevant substrate. The authors test the effects of mutations on binding to mononucleosomes, whereas their model is based on simultaneous binding of RYBP to one nucleosome and PRC1 to another. It seems likely that there are additional contacts with a di-nucleosome that do not form with single nucleosomes, for example with linker DNA. It is too much of a stretch to extrapolate from binding affinity measurements on a single nucleosome to the affinity of the complex for a di-nucleosome. In order to verify that the observed acidic patch contacts are indeed important, the experiments need to be done on a di-nucleosome or longer polynucleosome template. Protocols are available for assembling a di-nucleosome template in which one nucleosome contains H2A-Ub, which would be an appropriate substrate for testing the effects of mutations.

Our response: We understand the Reviewer's point and agree that studies in the context of longer polynucleosomes would be informative. Per the Reviewer's suggestion, we first attempted to assess binding of vPRC1 on unmodified di-nucleosomes by EMSA. However, we found the complex shift patterns in these EMSAs to be uninterpretable, and we were unable to draw any meaningful conclusions. We agree with the supposition made by the reviewer in that this is likely a result of different binding geometries vPRC1 possibly assumes on four different acidic patch surfaces. Things are further complicated by the observation that

for reasons not yet clear RYBP retains some binding activity, even if no Ubiquitin moiety is present on H2A (**Extended Data Figure 4b, c**), and that Ubiquitin on H2A does not interfere with the RING1B:BMI1-nucleosome interaction (revised **Figure 2a**). Given that we found it impossible to resolve the different bound species on unmodified di-nucleosomes, we have not pursued this further, but instead performed ubiquitination activity assays on oligonucleosome substrates as describe below.

Major point 2

Another issue is the reliance on binding, rather than activity assays. Since the model makes a prediction about activity, the effect of mutating the basic residues in RYBP should be tested on a di-nucleosome or polynucleosome template.

Our response: To assess activity, we performed ubiquitination assays with vPRC1 on short tetra-nucleosome arrays where we monitored the kinetics of H2Aub1 formation. One would expect that, in the array, upon monoubiquitination of H2A on a first nucleosome, the proposed read-write mechanism should facilitate monoubiquitination of linked nucleosomes and this effect should depend on interaction of the RYBP basic residues with the acidic patch. In our arrays, we found that wild-type vPRC1 showed greatly enhanced activity for H2Aub1 formation compared to the RING1B:BMI1 dimer alone. The ubiquitination activity of vPRC1 containing RYBP with mutations in the residues contacting the acidic patch was, however, greatly diminished compared to that of wild-type vPRC1 (**Figure 2c**). These activity assays provide strong support for the read-write mechanism proposed by our structural studies.

Major point 3

It seems that one prediction of the authors' model is that two vPRC1 complexes could bind to an unmodified di-nucleosome, whereas just a single complex would bind to a di-nucleosome (or multiply both by 2 if both faces are occupied). This could be a way to test the proposed competition model.

Our response: Unfortunately, as discussed in our response to Point 1 above, EMSAs have not allowed us resolve this issue due to the complications in using di-nucleosome substrates. We also believe that structural studies using vPRC1 and heterodimeric di-nucleosomes containing an unmodified nucleosome and a linked H2Aub1 modified nucleosome are beyond the scope of the present study.

Major point 4

In structure of vPRC1:NucH2Aub1, RYBP F32, I25 and T31 were identified to be essential for binding to Ub. Are there experiments (by the authors or in the literature) that verify the importance of these residues for either nucleosome binding or enzymatic activity?

Our response: A similar point was raised by reviewer #2. RYBP residues T31 and F32 were previously shown to be critical for the interaction of RYBP with free Ubiquitin (Arrigoni et al, 2006) and with H2Aub1 nucleosomes (Zhao et al, 2020). Specifically, the T31A/F32A double mutation in RYBP was previously reported to reduce RYBP binding to H2Aub1 nucleosomes in pull-down experiments (Zhao et al, 2020) and to impede monoubiquitination of unmodified H2A in nucleosomal arrays that contained pre-installed H2Aub1 on a subset of nucleosomes (Zhao et al, 2020). We performed EMSAs to measure the impact of these residues on RYBP alone or in context of vPRC1 binding to H2Aub1-modified mononucleosomes. The T31A/F32A

double mutation resulted in a reduction of the binding affinity compared to wild-type RYBP or wild-type vPRC1, and these experiments are presented in **Figure 2b** and **Extended Data Figure 5d**, respectively.

We also performed ubiquitination assays using vPRC1 containing RYBP_{T31A/F32A} on nucleosome arrays and found that the activity of this complex was reduced, but only during the early phases of the time course used in this experiment (**Figure 2c**). We note that we were unable to obtain a high-quality prep of an RYBP protein with a I25A/T31A/F32A triple mutation that may potentially result in a stronger phenotype because this mutant protein was only very poorly expressed.

Major point 5

As described in the Methods section on EM data processing for vPRC1:Nuc and vPRC1:NucH2Aub1, only ~2% (147,764/6,723,019) of particles and ~5% (640,334/12,150,042) of particles, respectively, were retained after 2D classification. Some justification or explanation should be given for the very small percentage of particles used to generate the 3D reconstruction.

Our response: Initial particle picking was done using Cryosparc's automated blob picker (for vPRC1:Nuc) or Relion's blob-based auto-picking (for vPRC1:NucH2Aub1). We opted for a low threshold to pick as many particles as possible. This strategy was based on the reasoning that streptavidin grids have a comparatively low particle contrast. While we did pick up more noise, the chances of discarding valuable particles were reduced. Consequently, many of the initial picks turned out to be background (~10,000,000 for vPRC1:Nuc and ~6,000,000 for vPRC1:NucH2Aub1), which were discarded in subsequent multiple rounds of 2D classification (as described in the Method section and as indicated in Supplementary figure 1 panel A and Supplementary figure 3 panel A). Particles were also excluded based on their visual appearance (very fuzzy 2D classes) or if they were low resolution (e.g. less than 7 Å). Given these considerations, the final set of particles that went into 3D ab-initio and make up the final model contained only ~2% (for vPRC1:Nuc) of particles and ~5% (vPRC1:NucH2Aub1) of the initial particles.

Major point 6

It is not clear why the structure of PRC1 bound to an unmodified nucleosome was reported given that there is already a higher resolution crystal structure of the complex. Some explanation should be given for including it.

Our response: There is an important difference between the structure reported here and the one reported from the crystal structure. The previously determined crystal structure contained the BMI1 Ring finger domain in complex with a fusion protein that consisted of the RING1B Ring finger domain linked to UBCH5C, bound to a mononucleosome (McGinty et al, 2014). In this case, crystallization was possible because UBCH5C made critical contacts to the DNA that stabilized the complex (McGinty et al, 2014). In our cryo-EM structure, we used full-length human RING1B, BMI1, and RYBP proteins. This difference provided interesting information about the mechanism used for binding, in that even in the absence of UBCH5C the RING1B:BMI1 dimer can still bind to the nucleosome acidic patch with a highly similar binding geometry and through the identical amino acid contacts. In **Extended Data Figure 2**, we have added images to illustrate this point showing the superposition of the two structures, i.e. that of RING1B:BMI1:RYBP bound to an unmodified nucleosome as determined by cryo-EM in our study and that of RING1B-UBCH5C:BMI1 bound to a nucleosome determined by X-ray crystallography (McGinty et al, 2014). Interpretation of both sets of data independently

informed our understanding of the molecular mechanism. Furthermore, our data serves as a 'control' for the structure of vPRC1 on H2Aub1 nucleosomes, where vPRC1 binds to H2Aub-containing nucleosomes preferentially via RYBP.

Major point 7

It would be helpful to readers to provide a diagram of the model for the read-write mechanism.

Our response: In **Figure 2d**, we now show a schematic model to illustrate the proposed read-write mechanism.

Minor point 1

The functional difference between the canonical and variant PRC1 complex should be described explicitly in the introduction.

Our response: In principle, we agree with your point that an extended introduction to discuss functional aspects of canonical and variant PRC1 and how they are linked to H3K27 methylation by PRC2 would be helpful. However, given the word limit of a Brief Communication in NSMB, we made the decision to allot more space to the presentation and discussion of the additional data that we now include in the revised manuscript. In an effort to still address this point, we now refer to a recent review, i.e. in lines 42-44 we wrote: "*For a comprehensive review about the composition and function of different forms of PRC1 and their role in the formation of Polycomb chromatin domains, see Blackledge and Klose (2021).*"

Minor point 2

In Figure 1g, 1e and 1h, protein names should be appropriately labeled for clearer presentation.

Our response: We have modified the panels in **Figure 1g-h** to include the protein names.

Minor point 3

In Page 8, line 172, vPRC1:NclH2Aub1 should be vPRC1:Nuch2Aub1.

Our response: We have corrected this.

Minor point 4

In the supplementary data, page 3, line 33, 3.5 A should be 3.5 Å.

Our response: We have corrected this.

Minor point 5

In page 17, line 372, the reference is not inserted properly.

Our response: We have corrected this.

Decision Letter, first revision:

Message: Our ref: NSMB-BC47936A

9th Jan 2024

Dear Dr. Mueller,

Thank you for submitting your revised manuscript "Structural basis of the histone ubiquitination read-write mechanism of RYBP-PRC1" (NSMB-BC47936A). It has now been seen by the original referees and their comments are below. The reviewers find that the paper has improved in revision, and therefore we'll be happy to accept it in principle in Nature Structural & Molecular Biology, pending revisions to satisfy the referees' final requests and to comply with our editorial and formatting guidelines.

To facilitate our work at this stage, it is important that we have a copy of the main text as a word file. If you could please send along a word version of this file as soon as possible, we would greatly appreciate it; please make sure to copy the NSMB account (cc'ed above).

Sincerely,

Dimitris Typas
Associate Editor
Nature Structural & Molecular Biology
ORCID: 0000-0002-8737-1319

Reviewer #1 (Remarks to the Author):

The authors have addressed the reviewers' concerns satisfactorily. I support publication.

Reviewer #2 (Remarks to the Author):

The authors have satisfyingly addressed all our comments. We recommend publication of this excellent work.

Reviewer #3 (Remarks to the Author):

The inclusion of activity data on tetranucleosome templates addresses one of the key issues raised in my review. Together with the additional biochemical data, I am satisfied that the authors have provided experimental support for their model. I have just one further comment, namely that the authors' explanation for including a small percentage of particles (major point 5) should be included (in abbreviated form) in the Methods section.

Overall, this is a very nice paper that will be of interest to the readership of NSMB.

Author Rebuttal, first revision:

Point-by-point response

Reviewer #3

"I have just one further comment, namely that the authors' explanation for including a small percentage of particles (major point 5) should be included (in abbreviated form) in the Methods section."

We have incorporated the explanation in the paragraph "Cryo-EM data processing" in the Methods section.

Final Decision Letter:

Message: 26th Feb 2024

Dear Dr. Mueller,

We are now happy to accept your revised paper "Structural basis of the histone ubiquitination read-write mechanism of RYBP-PRC1" for publication as a Brief Communication in Nature Structural & Molecular Biology.

Your paper will be published online soon after we receive proof corrections and will appear in print in the next available issue. You can find out your date of online publication by contacting the production team shortly after sending your proof corrections.

Please note that *Nature Structural & Molecular Biology* is a Transformative Journal (TJ). Authors may publish their research with us through the traditional subscription access route or make their paper immediately open access through payment of an article-processing charge (APC). Authors will not be required to make a final decision about access to their article until it has been accepted. Find out more about Transformative Journals

Sincerely,

Dimitris Typas
Associate Editor
Nature Structural & Molecular Biology
ORCID: 0000-0002-8737-1319